# Effects of Myofascial Release Technique along with Cognitive Behavior Therapy in University Students with Chronic Neck Pain and Forward Head Posture: A Randomized Clinical Trial

**DOI:** 10.3390/bs14030205

**Published:** 2024-03-04

**Authors:** Sahreen Anwar, Junaid Zahid, Cristina Ioana Alexe, Abdullah Ghazi, Gabriel Mareș, Zainab Sheraz, Rubén Sanchez-Gomez, Wajida Perveen, Dan Iulian Alexe, Qais Gasibat

**Affiliations:** 1Department of Physical Therapy, Lahore University of Biological and Applied Science, Lahore 53400, Pakistan; sahreen.anwar@lmdc.edu.pk; 2Department of Physical Therapy, Riphah International University, Islamabad 45320, Pakistan; theranajunaid786@gmail.com (J.Z.); abdullahghazi9999@gmail.com (A.G.); dr.zainab@riphahfsd.edu.pk (Z.S.); 3Department of Physical Education and Sports Performance, “Vasile Alecsandri” University of Bacău, 600115 Bacău, Romania; 4Department of Physical and Occupational Therapy, “Vasile Alecsandri” University of Bacău, 600115 Bacău, Romania; mares.gabriel@ub.ro; 5Nursing Department, Universidad Complutense de Madrid, 28040 Madrid, Spain; rusanc02@ucm.es; 6Instituto de Investigacion Sanitaria, (IDISSC) Hospital Clinico San Carlos, 28040 Madrid, Spain; 7School of Allied Health Sciences, CMH Lahore Medical College & IOD (NUMS), Lahore Cantt 54810, Pakistan; wajida_perveen@cmhlahore.edu.pk; 8Department of Physiotherapy, College of Medical Technology, Misrata P.O. Box 1458, Libya; gs57022@student.upm.edu.my

**Keywords:** chronic neck pain, cognitive behavioral therapy (CBT), forward head posture, myofascial release technique (MRT), neck disability

## Abstract

The purpose of this randomized controlled trial was to evaluate the effectiveness of the Myofascial Release Technique (MRT) along with Cognitive Behavioral Therapy (CBT) on pain, craniovertebral angle (CVA), and neck disability in university students with chronic neck pain and forward head posture. A total of sixty-six eligible participants with chronic neck pain and forward head posture were randomized into the Myofascial Release Therapy (MRT) group (*n* = 33) and MRT and Cognitive Behavior Therapy (CBT) group *(n* = 33). Clinical outcomes included neck pain measured using the numerical pain rating scale, neck disability measured through the neck disability index, and forward head posture measured through the cranial vertebral angle. The outcomes were assessed at baseline and the four and eight weeks after the intervention. Both groups showed significant improvement in pain intensity, CVA, and neck disability after the intervention. However, the CBT group demonstrated greater improvements than the MRT group. The difference in outcomes between the groups was statistically significant. Myofascial Release Therapy combined with CBT is an effective treatment method for patients with chronic neck pain and forward head posture.

## 1. Introduction

Neck pain is a highly prevalent musculoskeletal disorder that significantly impairs quality of life and interferes with social and professional activities. Neck discomfort is the fourth most common disability worldwide, with an annual prevalence ranging from 30% to 50%. Chronic neck pain is a discomfort that is felt anywhere along the posterior cervical spine from the superior nuchal line to the first thoracic spinous process and lasts for more than three months [1]. Students experience neck pain that not only affects their efficiency but also has a psychological and social impact on them and their families.

Chronic neck pain (CNP) sufferers have trouble keeping their heads up and leaning their heads slightly forward. A common postural aberration known as forward head posture (FHP) occurs when the head is positioned too far forward in relation to the body’s vertical axis, increasing the cervical spine’s lordosis [2]. The weight of the head of an adult is typically between 4 and 6 kg (8–12 pounds) in a neutral position, and forward head flexion at different angles can exert different forces on the cervical spine. For instance, when the head is flexed at a 15° angle, the neck is subjected to a strain of approximately 12 kg; at further angles such as 30°, 45°, and 60°, this force increases to 18 kg, 22 kg, and 27 kg, respectively [3]. During neck flexion, the cervical muscles not only oppose gravity’s pull to maintain good posture but also to prevent neck strain and injury [4]. The neck, shoulders, and upper back muscles may become strained during a forward head posture because the head is inappropriately supported by the spine [5].

Due to contemporary trends and an increased dependence on technology, numerous people are struggling with CNP and FHP [6]. This issue is well-documented in the literature and is a matter of great concern for ergonomists and occupational health professionals [7]. Various studies suggest that a significant number of university students (63.96%) exhibit forward head posture (FHP), which may be a result of excessive gadget usage and long sitting hours in inappropriate positions during lectures and labs [8]. A state of balanced musculoskeletal alignment that promotes good posture is believed to reduce stress and strain on the spine and related structures.

Many treatment strategies have been implemented worldwide for the management of CNP and correction of FHP [9,10,11,12]. Apart from medications and exercise therapy, a biopsychosocial approach may be an effective strategy for minimizing the disability caused by chronic neck pain and poor posture. Evidence suggests that strength training of the neck and surrounding muscles may be useful in reducing neck discomfort and related disabilities. Static exercises, commonly referred to as isometric exercises, can be an efficient technique for developing weak muscles without putting pain-sensitive tissues such as tendons, ligaments, or neck joints through unnecessary agony.

Myofascial trigger points (TrPs), which appear as sensitive areas in the palpable taut bands of skeletal muscles, are frequently present in patients with CNP. These trigger points can cause pain and discomfort in the affected area. Myofascial trigger points (TrPs) can have a variety of effects, including altered muscle activation, restricted range of motion, exhaustion, elevated muscle tension, and autonomic abnormalities. These effects are typically characteristic of trigger point pain and can cause significant discomfort. Simon’s criteria for assessing trigger points are widely used by healthcare professionals. The criteria include an elongated, tight band in the muscle, a palpable hypersensitive spot in the muscle, reproduction of the patient’s pain when the trigger point is palpated, and a restricted range of motion and muscle weakness associated with the affected muscle [13]. Generally, people with neck discomfort typically prefer self-care strategies to manage their symptoms. According to recent studies, myofascial release is highly effective in treating forward head posture, extending the range of motion for side flexion and rotation, and enhancing the general quality of life [14].

Chronic neck pain is strongly linked to poor psychological health, including cognitive discomfort, anxiety, and depression [15]. If left untreated, this condition can lead to a cycle of avoidance, impairment, and increased pain, worsening psychological distress. Cognitive therapy, commonly referred to as cognitive behavior therapy (CBT), is a treatment strategy for treating the psychological and behavioral effects of chronic pain. This therapy focuses on modifying behaviors and thought patterns that are believed to contribute to or exacerbate the problem. Cognitive therapy improves the patient’s capacity to function despite the presence of pain rather than lowering the pain itself. It helps patients identify and manage environmental triggers that worsen their pain, and they learn to identify chronic pain and the resulting inability as something tolerable through cognitive therapy. According to a study, cognitive therapy can help individuals with chronic pain to live comfortably by altering their behavior in response to pain [16]. In addition, adding cognitive-behavioral therapy to treatment can lead to better outcomes and improve patients’ quality of life [17].

There is a lack of evidence regarding a multimodal approach for addressing the physical and psychosocial aspects of CNP with FHP. A bio-psychosocial approach is essential for treating the biological, psychological, and social factors associated with chronic neck pain [18].

The objective of the present study was to evaluate the effectiveness of the Myofascial Release Technique (MRT) along with Cognitive Behavioral Therapy (CBT) on craniovertebral angle (CVA), pain, and neck disability in university students with CNP and FHP.

## 2. Materials and Methods

### 2.1. Study Design

This randomized, single blinded (patient blinded), controlled parallel trial, with a 1:1 allocation ratio between the two groups, was prospectively registered (IRCT20230216057434N2) on the Iranian Registry of Clinical Trials. The trial was conducted in accordance with the principles of the Consolidated Standards of Reporting Trials (CONSORT) [19]. The details of participant recruitment are given in Figure 1. Study was approved by the ethical committee of Riphah International University IRB-RCRAHS-REC/23/07. The recruitment of the participants started on 6 March 2023 and was completed on 14 June 2023. Data were gathered from the outpatient Physiotherapy Department of Safi Teaching Hospital, Riphah International University, Faisalabad, Pakistan.

### 2.2. Sample Size Calculation

The sample size was calculated using the Giga Calculator with 80% study power, 5% margin of error, and 95% confidence interval, and 60 participants were found. By considering probable attrition during the study, a 10% attrition rate was added, and the final sample size was 66.

### 2.3. Inclusion and Exclusion Criteria

University students (Male and female) between the ages of 18 and 28 years [20] with grade 2 neck pain (as per the Koninklijk Nederlands Genootschap Fysiotherapie (KNGF) Guidelines) [21] for three or more consecutive months] and a craniovertebral angle (CVA) of less than 53 degrees [22], were included in the study. Students with a history of trauma, spinal surgery, congenital deformities, fibromyalgia, and cervical radiculopathy were excluded from the study. Furthermore, students who received pain management injections for trigger points were excluded from the study [20].

### 2.4. Randomization and Masking

Randomization was performed using computer software to allocate patients to the MRT and CBT groups. The “Random Allocation Software Version 2” was used for randomization. A total sample size of 66 was entered into the software to equally allocate the patients into two groups.

The group allocation was kept concealed; the study employed a single-blind approach, ensuring that all patients were unaware of the randomization process and remained blinded to the type of intervention being administered throughout the study. The random allocation sequence was generated by a designated computer operator, participants were enrolled in the study by a front desk officer, and participants were assigned to the intervention groups by another staff member who did not directly execute the procedures. The interventions were performed by designated physiotherapists who were trained to carry out the procedures beforehand. Data were gathered at each step in line with the study protocol.

### 2.5. Intervention

To start the session, the students in the cognitive behavior therapy group adopted a prone position, and a hot pack was applied to the cervical region for five minutes. Natural wax was used to ensure the smoothness of strokes and prevent the rubbing of skin. All three layers of superficial fascia, deep fascia, and myofascial interface were engaged, and the sequence of the techniques moved from the superficial layer to deeper layers gradually. Once the exact location of pain was identified by palpation, MFR was applied in the following sequence: First, skin rolling, a superficial stroke technique, was performed for 2–3 min on the neck and shoulders area to promote general relaxation. The therapist then concentrated on the troublesome area and applied a Myofascial Release Technique, stretching the deep fascia in a transverse and reciprocating way using cross-hand stretch and varying the pressure according to the student’s tolerance for pain. The therapy session ended with another 2–3 min of surface stroke massage. The Myofascial Release Technique was followed by neck isometric exercises in all six ranges of cervical flexion: right- and left-sided flexion-extension, and rotation. A 10 s hold was improvised for each isometric contraction, and 10 contractions were performed in each direction. After completing a set of these six movements, the set was repeated five times, with a 5 s rest period between each repetition.

This treatment session was followed by Cognitive Behavioral therapy for 20 min using visual aids such as videos, images, animations, and simple pamphlets and texts to explain the cervical spine’s structure and biomechanical behavior to preserve optimal ergonomics. The concepts of pain, pain pathways, and sensory input leading to motor feedback and techniques to control flare-ups and manage pain by maintaining good posture during activities of daily living were also explained. It was ensured that the students were actively engaged and focused throughout the session. Additionally, the physiotherapist provided the students with a manual of information (booklet) highlighting the key points presented throughout the informative session. The students were also encouraged to ask any questions they had. The entire treatment session was approximately 45 min long.

The Myofascial Release Therapy (MRT) group received hot packs and MRT and neck isometric exercises were similar to those performed in the CBT group. After completing the session, a reference book on ergonomics and design was given to the students [23]. The students were instructed to read the recommendation for 20 min. Therefore, the entire treatment session was approximately 45 min long. Both groups underwent two sessions per week for 8 consecutive weeks.

### 2.6. Outcome Measures

Numeric Pain Rating Scale (NPRS) was used to assess pain intensity [24]. An Urdu version of the Neck Disability Index (NDI-U) was used to assess neck-specific disability. The questionnaire consisted of 11 items measuring neck-related disability and its impact on daily activities [25]. FHP was measured using the cranial vertebral angle measured using the Photogrammetric Method with Kinova software version 0.8.27-64 bit, Kinova Corporation Canada (Boisbriand, QC, Canada) for cervical range of motion in the sagittal plane (Figure 2). The photogrammetry approach has been successfully used in various studies and is considered reliable for assessing forward head posture [26,27].

The details of the measurement of the craniovertebral angle are given below:

The students stood at a specific distance from the wall, with their left side facing everyone. Two anatomical landmarks, the left tragus (near the ear) and the spinous process of the C7 vertebra (at the base of the neck) were marked for angle calculation. The camera was positioned at a distance of 265 cm and adjusted to the student’s right shoulder height using a photographic tripod. Students were asked to perform specific movements, such as raising their hands and leaning forward, to establish a natural posture. After a brief pause, lateral-view photographs were taken. The angle formed by connecting the marked points with horizontal and vertical lines was measured using Kinova software version version 0.8.27-64 bit, Kinova Corporation Canada This angle represents the forward head posture (Figure 2) [26].

### 2.7. Data Analysis

Data analyses were conducted using SPSS 23.0 (SPSS Inc., Chicago, IL, USA). Normality of the data was checked using Kolmogorov–Smirnov test. As the data were found to be normally distributed (*p* > 0.05), parametric tests were applied to compute the results. For descriptive statistics, continuous variables are expressed as mean ± SD (standard deviation). For the inferential statistics, mixed method analysis of variance (MANOVA) was conducted for between- and inter-group comparison among the MRT and CBT exercise groups at baseline, the 4th week, and the 8th week. For pair-wise comparison of variables at baseline, 4th- and at 8th-week paired sample *t* tests were used. A significance level of *p* ≤ 0.05 was considered statistically significant with a 95% confidence interval.

## 3. Results

Of the 100 participants recruited for the study, sixty-six fulfilled the inclusion criteria and were randomly assigned to the MRT and CBT groups. In total, seven patients were lost to follow-up for different reasons, four from the CBT group and three from the MRT group. Fifty-nine patients were analyzed in the eighth week of the intervention. There were no significant differences between the groups in terms of mean age, BMI, pain, cranial vertebral angle, and neck disability at baseline (*p* > 0.05). The average age, number of male and female participants, and baseline measurements for the MRT and CBT groups are given in Table 1.

The results of the MIXED METHOD analysis of variance MANOVA for the evaluation of changes in mean scores for pain over time (from base line to the fourth and eighth weeks) revealed a significant group-by-time interaction (F = 29.338); *p* ˂ 0.001) (Table 2). At baseline, the mean pain scores in the MRT and CBT group were 4.757 ± 1.658 and 5.606 ± 1.748, respectively. Compared to the corresponding baseline values, the improvement in the CBT group was significant compared with the MRT group in the fourth week, changing from F = 4.089, *p* = 0.047 in the fourth week to and F = 29.338, *p* ≤ 0.001 in the eighth week (Table 2).

The between-groups comparison of the MRT and CBT groups showed mean ±SD values for the craniovertebral angle at baseline of (40.818 ± 5.826) and (40.843 ± 5.558), respectively. After the fourth week of intervention, the mean CVA angle was improved to (46.106 ± 5.928) for CBT group and (44.710 ± 5.698) for the MRT group. After eight weeks of intervention, the mean ± SD values for the craniovertebral angle were (49.655 ± 6.545) for the CBT group and (45.456 ± 6.821) for the MRT group, with a statistically significant difference of *p* = 0.019 (Table 3).

The between-groups comparison of the CBT and MRT groups at baseline showed mean ± SD values for the neck disability index of (18.060 ± 7.777) and (15.878 ± 7.801), respectively, with a *p*-value of (0.259). The mean± SD values were improved to (11.366 ± 3.952) and (9.322 ± 4.407) for the CBT and MRT groups, respectively, in the fourth week of the intervention with a *p*-value of (0.062). In the eighth week of the intervention, the mean± SD values for the NDI further improved to (5.551 ± 2.180) and (7.166 ± 3.130) for the CBT and MRT groups, respectively, with a significant *p*-value of (0.026) (Table 4).

These findings highlight the potential benefits of integrating psychological therapy, such as CBT, alongside physical interventions, such as MRT, in the treatment of chronic neck pain and forward head posture. The results provide valuable insights for healthcare professionals involved in treating this patient population, emphasizing the importance of a comprehensive and multidimensional approach to addressing both the physical and psychological factors contributing to chronic neck pain and postural problems.

A pairwise comparison of the NPRS, CVA, and NDI variables at baseline, in the fourth week, and the eighth week is shown in (Table 5). It revealed significant differences in the CBT group in the fourth week and eighth week as compared to the baseline values suggesting that it is an effective treatment strategy.

## 4. Discussion

The findings of the current study demonstrated that the CBT + MRT group resulted in significant improvements in pain (*p* = 0.001), neck disability (*p* = 0.026), and craniovertebral angle (*p* = 0.019) compared with the only MRT group, demonstrating a statistically significant difference between the groups. However, the combination of MRT with CBT demonstrated significant outcomes compared with MRT alone, indicating that incorporating CBT into treatment approaches enhanced the effectiveness of the intervention.

The greater improvements observed in the MRT combined with CBT group could be attributed to several factors. First, the inclusion of CBT in the treatment protocol targeted psychological factors such as maladaptive thoughts and behaviors. By addressing these psychological aspects, participants in the combined intervention group may have experienced enhanced coping strategies, increased self-efficacy, and improved self management of their condition. These psychological factors may have contributed to better treatment outcomes in terms of pain reduction and functional improvements. Furthermore, CBT may have facilitated behavioral changes and increased adherence to postural modifications and ergonomic recommendations, which could have positively influenced the craniovertebral angle and postural alignment. The cognitive restructuring techniques employed in the CBT sessions may have helped participants to recognize and modify harmful postural habits and maintain proper alignment throughout their daily activities.

According to a secondary study, CBT in combination with other treatments was found to be successful in alleviating neck pain in patients with chronic neck pain [28]. The findings of the current study support the growing recognition of the importance of using a multidimensional approach to manage chronic neck pain and postural problems. By combining physical interventions, such as MRT, with psychological strategies, such as CBT, a more comprehensive treatment approach is provided which addresses both the physical and psychological aspects of the condition.

The findings of our study correlate with the results of a previous study conducted by Urits et al., which showed CBT to be a beneficial intervention in managing pain associated with cervicogenic headaches and fibromyalgia. The positive effects of CBT may be attributed to its ability to reduce activity in the posterior cingulate cortex, leading to a reduction in pain intensity and improvements in pain-related cognition and anxiety. In addition, CBT provides chronic pain patients with improved cognitive skills and offers increased accessibility to those who may have difficulties attending in-person therapy sessions [29]. The findings of the current study demonstrated that the CBT + MRT group resulted in significant improvements in pain (*p* = 0.001), neck disability (*p* = 0.026), and craniovertebral angle (*p* = 0.019) compared with the MRT group, demonstrating a statistically significant difference between the groups.

In a systematic review, CBT was found to have significant effects on pain relief for subacute and CNP. However, when compared with other interventions, CBT showed significant effects on pain relief and non-significant effects on disability. The overall quality of the evidence was deemed to be low and the observed effects were not considered clinically meaningful [30]. In contrast to the review, the current study’s findings demonstrated that the CBT + MRT group showed improvements in neck disability (*p* = 0.026) as compared to MRT group, demonstrating a statistically significant difference between the groups.

Various studies have been conducted in which the effects of Myofascial Release Therapy were assessed in comparison to other clinical measures in chronic neck pain management. The results of this study are in line with those of a previous study in which it was concluded that it is an effective treatment strategy for managing pain and disability [31,32]. In another study, MRT combined with cryostretching demonstrated significant results in terms of improving pain and range of motion. However, there was no explanation of how long the effects of these treatment strategies last [33]. The current study’s findings demonstrated that the CBT group resulted in statistically significant differences in pain (*p* < 0.001), neck disability (*p* = 0.026), and craniovertebral angle (*p* = 0.019) as compared to the MRT group. These consistent findings support the notion that Myofascial Release combined with CBT might be a beneficial treatment option for individuals experiencing non-specific chronic neck pain. This integrative approach aligns with the bio-psychosocial model of care, acknowledging the interplay among biological, psychological, and social factors in chronic pain.

### 4.1. Limitations of the Study

The study may have been limited by the comparatively small sample size, which could affect the generalizability of the results to larger populations. This study focused specifically on university students with CNP and FHP. Therefore, the findings may not be directly applicable to other populations or individuals with different characteristics.

The duration of the intervention was relatively short; a longer duration could provide a more comprehensive understanding of the long-term effects of MRT and Cognitive Behavioral Therapy. Relying on self-reported measures, such as pain levels and disability, introduces the potential for bias and a subjective interpretation of the results. No long-term follow-up was included to assess the sustainability of the observed improvements.

### 4.2. Strength and Implications of the Study

The strength of the study lies in the fact that the physical and behavioral aspects of neck pain have been targeted using CBT and the intervention not only has long term effects but enhances the treatment’s efficacy as well.

The target intervention may be studied further with other practiced interventions and in different populations to establish evidence about its efficacy and outcomes.

## 5. Conclusions

The present study provides evidence for the effectiveness of the combination of the Myofascial Release Technique (MRT) with Cognitive Behavioral Therapy (CBT) in managing CNP and improving postural alignment in university students. The incorporation of CBT into treatment approaches demonstrated greater improvements in pain intensity, craniovertebral angle (CVA), and neck disability than MRT alone. These findings highlight the potential benefits of integrating psychological strategies alongside physical interventions, supporting the importance of taking a comprehensive and multidimensional approach to managing chronic neck pain and postural problems. Further studies are needed to explore the long-term effects and generalizability of these findings.

## Figures and Tables

**Figure 1 behavsci-14-00205-f001:**
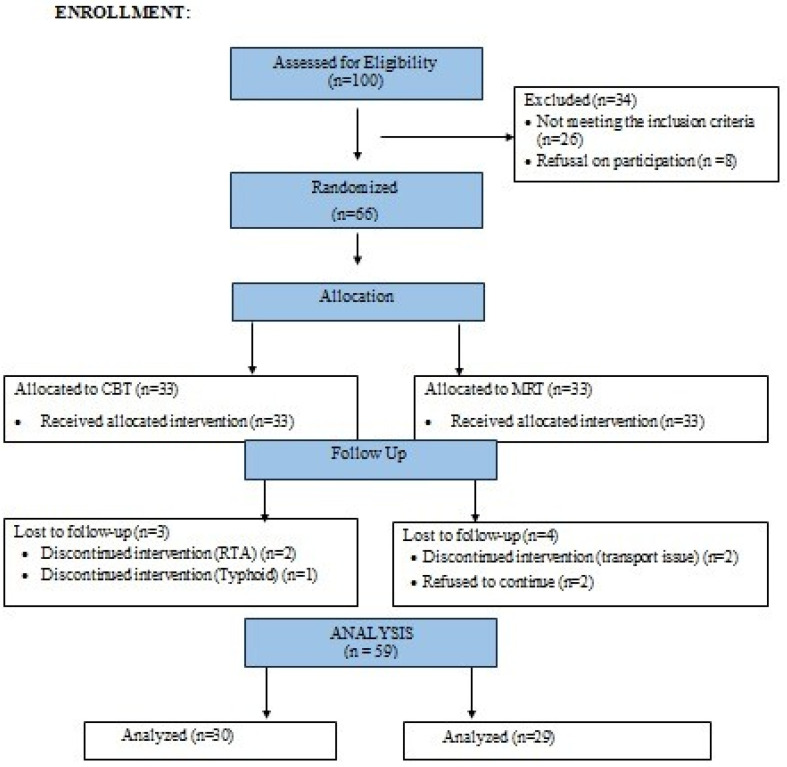
Consort Diagram.

**Figure 2 behavsci-14-00205-f002:**
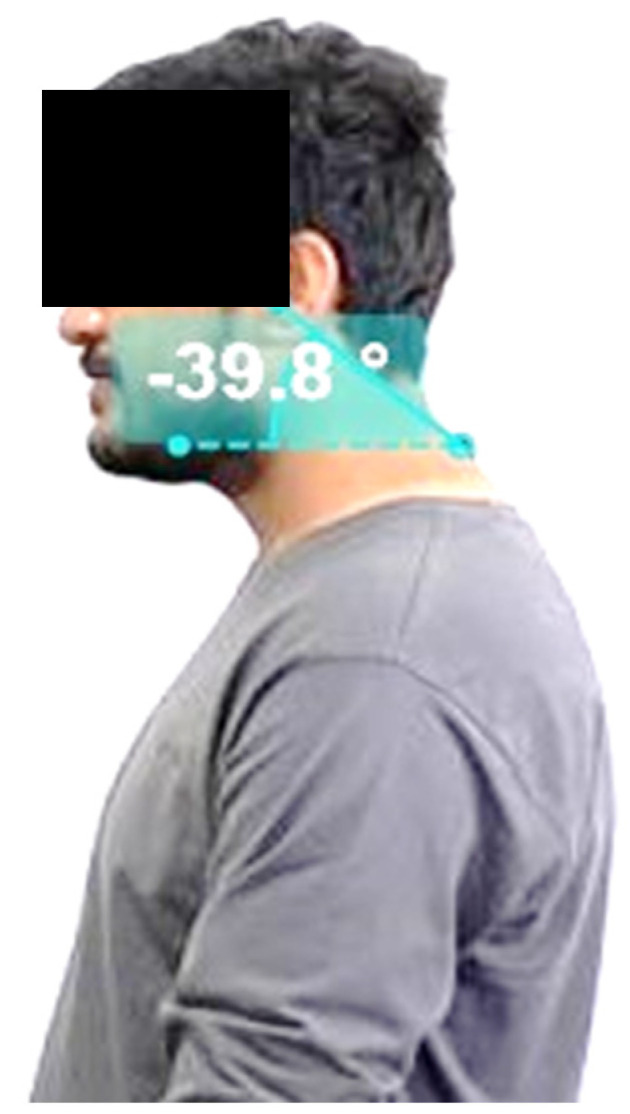
Measurement of the Craniovertebral Angle.

**Table 1 behavsci-14-00205-t001:** Baseline descriptive characteristics.

	CBT	MRT
Outcome Measures	Mean ± SD	Mean ± SD
Gender	M = 24	M = 24
	F = 09	F = 09
Age (Years)	22.6364 ± 1.76455	22.1212 ± 1.57634
NPRS at baseline	5.6061 ± 1.74892	4.7576 ± 1.65888
CVA angle at baseline (Degrees)	40.8433 ± 5.55882	40.8188 ± 5.82644
NDI at baseline	18.0606 ± 7.77793	15.8788 ± 7.80127

SD = standard deviation; NPRS = numeric pain rating scale; CVA = craniovertebral angle; NDI = neck disability index.

**Table 2 behavsci-14-00205-t002:** Between-group comparison of NPRS.

Outcome Measures		N	Mean	Std. Deviation	95% Confidence Interval for the Mean		F	*p* Value
					Lower Bound	Upper Bound		
NPRS at baseline	MRT and CBT	33	5.606	1.748	4.985	6.226	1.910	0.172
	MRT Alone	33	4.757	1.658	4.169	5.345		
NPRS in the fourth week	MRT and CBT	30	3.133	1.041	2.744	3.522	4.089	0.047
	MRT Alone	31	3.548	1.286	3.076	4.020		
NPRS in the eighth week	MRT and CBT	29	1.241	0.510	1.047	1.435	29.338	<0.001
	MRT Alone	30	2.366	0.999	1.993	2.739		

NPRS: Numeric pain rating scale.

**Table 3 behavsci-14-00205-t003:** Between-group comparison of craniovertebral angle.

Outcome Measures		N	Mean	Std. Deviation	95% Confidence Interval for the Mean		F	*p* Value
					Lower Bound	Upper Bound		
CVA sngle at the Baseline	MRT and CBT	30	40.843	5.558	38.767	42.919	0.000	0.987
	MRT	32	40.818	5.826	38.718	42.919		
CVA angle in the fourth week	MRT and CBT	29	46.106	5.928	43.852	48.361	0.852	0.360
	MRT	30	44.710	5.698	42.582	46.837		
CVA angle in the eighth week	MRT and CBT	29	49.655	6.545	47.165	52.145	5.812	0.019
	MRT	30	45.456	6.821	42.909	48.003		

**Table 4 behavsci-14-00205-t004:** Between-group comparison of neck disability index (NDI).

Outcome Measures		N	Mean	Std. Deviation	95% Confidence Interval for the Mean		F	*p* Value
					Lower Bound	Upper Bound		
NDI at baseline	MRT and CBT	33	18.060	7.777	15.302	20.818	1.294	0.259
	MRT Alone	33	15.878	7.801	13.112	18.645		
NDI after the fourth week	MRT and CBT	30	11.366	3.952	9.890	12.842	3.629	0.062
	MRT Alone	31	9.322	4.407	7.705	10.939		
NDI after the eighth week	MRT and CBT	29	5.551	2.180	4.722	6.381	5.253	0.026
	MRT Alone	30	7.166	3.130	5.997	8.335		

**Table 5 behavsci-14-00205-t005:** Pairwise comparison of pain, CVA, and NDI at baseline, fourth week, and eighth week.

Outcome Measures	Combinations at Different Time Points	Mean Difference	*p*-Value	95% Confidence Interval for the Difference	
				Lower Bound	Upper Bound
NPRS	Baseline to Fourth Week	−2.509	<0.001 *	−2.825	−2.193
	Baseline to Eighth Week	−4.682	<0.001 *	−5.203	−4.161
	Fourth Week to Eighth Week	−2.173	<0.001 *	−2.558	−1.788
Craniovertebral angle (degrees)	Baseline to Fourth Week	−2.463	<0.001 *	−2.798	−2.193
	Baseline to Eighth Week	−5.193	<0.001 *	−5.745	−4.647
	Fourth Week to Eighth Week	−2.731	<0.001 *	−3.172	−2.298
Neck Disability Index	Baseline to Fourth Week	−2.323	<0.001 *	−2.627	−2.019
	Baseline to Eighth Week	−4.088	<0.001 *	−4.476	−3.701
	Fourth Week to Eighth Week	−1.766	<0.001 *	−1.987	−1.544

NPRS: Numeric pain rating scale. “*” indicates the statistically significant results

## Data Availability

Data can be provided following a reasonable request to the first author. The protocol of the study is available on the Iranian register of clinical trials.

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
