# Peer review of "Effects of Myofascial Release Technique along with Cognitive Behavior Therapy in University Students with Chronic Neck Pain and Forward Head Posture: A Randomized Clinical Trial"

_behavsci, 2024, doi:10.3390/bs14030205_

Round 1
Reviewer 1 Report
Comments and Suggestions for Authors
Dear. Author,
Thank you for submitting your manuscript. This randomized controlled study is well-designed in its aims and methods and will likely interest relevant clinicians. However, a few minor corrections need to be made before the paper is published.
1. In the introduction, I recommend unifying the units when referring to the loads on the skull due to head posture.
2. In the Results area, please indicate which table you are describing in the body of the text, even though it is well organized.
3. The body of the table at the bottom of Table 3 is given subheadings not found in the other result areas. Please keep the format of your text consistent.
4. In the Results section, there is a missing explanation for Table 5.
5. In the 6th and 7th paragraphs of the Discussion section, the authors argue for the utility of applying MRT for chronic cervical pain; however, there is no mention in the Results section that MRT explains the improvement in pain.
6. In the Discussion section, paragraphs #6-9 describe the current study and previous research. Structurally, it is a series of unnecessary descriptions, so it is recommended that they be reorganized for clarity.
Thank you.
Comments on the Quality of English LanguageThe English language of the Complaint is considered to be reasonably plain.
Author Response
The authors thank reviewer 1 for the support, opinions expressed, advice and time given to study our article. Time is important and for this aspect we are grateful to the reviewer.
Please see the attachment

Reviewer 2 Report
Comments and Suggestions for Authors
It is an interesting and valuable RCT. I sincerely congratulate you for your work.
Comments and suggestions:
1) In ABSTRACT, it is not well defined that the CBT group also includes MRT treatment, in the paragraph: "were randomized into Myofascial Release Therapy (MRT) group (n=33) and Cognitive Behavior Therapy (CBT) group (n =33)".
I recommend putting "MRT and Cognitive Behavior Therapy (CBT) group (n=33)"
2) In the Introduction, express the progression in Kg according to the degrees of craniocervical inclination; Kg being the preferable measurement, as it belongs to the International System. But first express the weight of the head in POUNDS. I recommend putting the Kg corresponding to those 10 to 12 Pounds in parentheses.
3) I think that where it says in the text Fig. 1, it refers to Fig. 2
4) The graphic quality of Figure 2 is poor. I recommend improving it, for greater graphic quality of the article.
5) In 2.5 INTERVENTION: "The therapist then concentrated on the troublesome area and applied a myofascial release technique, varying the pressure according to the student's tolerance for pain." I consider it important to describe in more detail how they performed the myofascial release technique, to know their protocol well and to make it reproducible in the clinic and future research.
6) In DISCUSSION, although it is clear in the first paragraph, to more clearly express the interventions in both groups, in the paragraph: The findings of current study demonstrated that the CBT group resulted in significant improvements in pain (p=0.001), neck disability (p=0.026), and craniovertebral angle (p=0.019) compared with the MRT group, demonstrating a statistically significant difference between the groups. In my opinion, it is better if it says: "The findings of current study demonstrated that the CBT+MRT group resulted in significant improvements in pain (p=0.001), neck disability (p=0.026), and craniovertebral angle (p=0.019) compared with the only MRT group, demonstrating a statistically significant difference between the groups", or similar. The same in other successive paragraphs of the discussion.
​
Author Response
The authors thank reviewer 2 for the support, opinions expressed, advice and time given to study our article. Time is important and for this aspect we are grateful to the reviewer.
Please see the attachment
